# Transformation of *Dunaliella salina* by *Agrobacterium tumefaciens* for the Expression of the Hemagglutinin of Avian Influenza Virus H5

**DOI:** 10.3390/microorganisms10020361

**Published:** 2022-02-04

**Authors:** Inkar Castellanos-Huerta, Gabriela Gómez-Verduzco, Guillermo Tellez-Isaias, Guadalupe Ayora-Talavera, Bernardo Bañuelos-Hernández, Víctor Manuel Petrone-García, Gilberto Velázquez-Juárez, Isidro Fernández-Siurob

**Affiliations:** 1Programa de Maestría y Doctorado en Ciencias de la Producción y de la Salud Animal, Facultad de Medicina Veterinaria y Zootecnia, Universidad Nacional Autónoma de México, Ciudad Universitaria, Ciudad de Mexico 04510, Mexico; 2Departamento de Medicina y Zootecnia de Aves, Facultad de Medicina Veterinaria y Zootecnia, Universidad Nacional Autónoma de México, Avenida Universidad 3000, Ciudad de Mexico 04510, Mexico; gagove@unam.mx; 3Department of Poultry Science, University of Arkansas, Fayetteville, AK 72701, USA; gtellez@uark.edu; 4Centro de Investigaciones Regionales, Dr. Hideyo Noguchi, Universidad Autonoma de Yucatán (UADY), Mérida 97000, Mexico; talavera@correo.uady.mx; 5Escuela de Veterinaria, Universidad De La Salle Bajío, Avenida Universidad 602, Lomas del Campestre, León 37150, Mexico; berbanher@gmail.com; 6Departamento de Ciencias Pecuarias, Facultad de Estudios Superiores Cuautitlán UNAM, Cuautitlán Izcalli 54714, Mexico; vmpetrone@hotmail.com; 7Departamento de Química, Centro Universitario de Ciencias Exactas e Ingenierías, Universidad de Guadalajara, Blvd. Marcelino García Barragán #1421, Guadalajara 44430, Mexico; gilberto.velazquez@academicos.udg.mx; 8Viren SA de CV, Presidente Benito Juárez 110B, José María Arteaga, Querétaro 76135, Mexico; ifsviren@gmail.com

**Keywords:** avian influenza, *Dunaliella salina*, *Agrobacterium tumefaciens*, hemagglutinin, recombinant

## Abstract

Avian influenza (AI) is one of the main threats to the poultry industry worldwide. Vaccination efforts are based on inactivated, live attenuated, and recombinant vaccines, where the virus hemagglutinin (HA) is the main component of any vaccine formulation. This study uses *Dunaliella salina* to express the AIV HA protein of an H5 virus. *D. salina* offers a system of feasible culture properties, generally recognized as safe for humans (GRAS), with N-glycosylation and nuclear transformation by *Agrobacterium tumefaciens*. The cloning and transformation of *D. salina* cells with the H5HA gene was confirmed by polymerase chain reaction (PCR). SDS-PAGE and Western blot confirmed HA5r protein expression, and the correct expression and biological activity of the HA5r protein were confirmed by a hemagglutination assay (HA). This study proves the feasibility of using a different biological system for expressing complex antigens from viruses. These findings suggest that a complex protein such as HA5r from AIV (H5N2) can be successfully expressed in *D. salina*.

## 1. Introduction

Avian influenza (AI) is one of the most critical illnesses in the poultry industry worldwide [1]. AI is defined as a systemic disease ranging from clinically undetected to severe with a high mortality depending on the virus subtype [2]. The AI virus (AIV), a member of the *Orthomyxoviridae* family, is the etiologic agent of AI [3]. Hemagglutinin (HA) and neuraminidase (NA) are the major glycoproteins found on the surface of the influenza virus [4], though HA is the most abundant (up to 10-fold more than NA) [5]. HA is a trimeric elongated rod-shaped protein with three monomers (75 kDa) connected in a trimer (225 kDa) with a length of 10–14 nm and a 4–6 nm diameter. Two polypeptides (HA1 and HA2) are organized into a core-helical coiled-coil (stem-like domain) with N-linked oligosaccharide side chains and three globular heads in the monomeric structure of HA (HA0) [6]. The primary role of HA is to act as a viral receptor by interacting with sialic acid (SA) to allow the virus to bind and attach to the host cells [6,7]. Influenza virus HA is the primary inducer of host immunity; thus, vaccination is the most crucial tactic for combating AIV on the ground and is mostly focused on the HA subtype in circulation [8,9]. Currently, the expression of HA in heterologous systems demonstrates the feasibility of its use as a strategy against AIV [6,8,9,10,11,12]. Among the recombinant protein expression systems, particular characteristics can be described. These examples include high production costs in the case of animal and insect cells, undesirable post-translational modifications observed in yeasts, prolonged cultivation times as in plants, and the lack of glycosylation in proteins as in the case of bacteria models [13]. However, there are other systems, such as microalgae, where their use has not been fully explored. A potential system of expression proposed is the eukaryotic green alga, *Dunaliella salina* [14].

*Dunaliella sp.* is a unicellular, halophilic, bi-flagellate, and naked green alga, Phylum *Chlorophyta*, Class *Chlorophyceae*, order *Volvocales*, family *Polyblepharidaceae*, and genus *Dunaliella* with a total of 29 species [15]. It was first described in 1905 [16] and named in honor of Michel Felix Dunal [17]. Currently, the genetic manipulation of *Dunaliella sp.* includes electroporation [18], particle bombardment [19], glass beads [20], lithium acetate/polyethylene glycol (PEG)-mediated [21], and genetic nuclear transformation by *Agrobacterium tumefaciens* [22,23], which present different degrees of effectiveness for the expression of heterologous genes [24]. The *A. tumefaciens*-mediated nuclear transformation system has demonstrated the stable integration of recombinant genes into host cells [25], taking advantage of the capacity of *A. tumefaciens* to transfer foreign genes to the host genome using the T- DNA region from a binary vector [26,27].

The *D. salina* system for antigen production offers industrial advantages such as low production costs, low risk of biological contamination with animal pathogens, the capacity for post-translational modification such as N-glycosylation in proteins similar to human glycoproteins, and the possible oral administration of the antigen [28]. This study aimed to establish the expression of the recombinant H5rD protein of AIV in the nucleus of *D. salina* and examine its bioactivity in vitro.

## 2. Materials and Methods

### 2.1. Synthetic Design of the H5rD Gene

The synthetic design of the *H5rD* gene was based on the whole sequence of the HA gene of the reference strain A/chicken/Hidalgo/28159-232/1994 (H5N2) Genbank # CY006040.1, which consists of 1695 bp and presents a low-pathogenicity cleavage site [29]. The gene *H5rD* was synthesized and codon-optimized for *D. salina* by GenScript Inc. (GenScript, Piscataway, NJ, USA).

### 2.2. D. salina Strain and Culture Conditions

The UTEX-1644 strain of *D. salina* from the Culture Collection of Algae at the University of Texas (Austin, TX, USA) was purchased and used for all the experiments. *D. salina* was cultured in PKS (phosphate-potassium-sodium) modified medium at 26 °C over 12-h night–day cycles under a constant light intensity (30 μmol photons m^−2^ s^−1^) provided by fluorescent lamps [30]. At the logarithmic growth phase (10^5^ cells mL^−1^), cells were harvested, centrifuged, and analyzed for their size uniformity, shape, and movement during the complete experiment.

### 2.3. DNA Cloning Vector pH5HPDS

A DNA cassette of expression (2493 bp) contained two promoters upstream: heat shock protein 70 (Hsp70) (275 bp) and RUBISCO small subunit (199 bp). Downstream, it contained an RBCS2 terminator (hybrid promoter/terminator sequences strategy) (234 bp) [24,31] of the cloning site for the *H5rD* exogenous gene. The complete cassette was synthesized by GenScript Inc. (GenScript, Piscataway, NJ, USA) and subcloned into the binary vector pCAMBIA-1301 (Genbank No.AF234297) at the *EcoRI-PstI* sites by standard cloning methods [32]. The resulting vector named pH5HPDS was confirmed by restriction analysis to verify the correct integrity. The vector pH5HPDS was transferred into *Agrobacterium tumefaciens* (LBA4404 strain) by electroporation [33]. The sequence and map of the constructed vector pH5HPDS are available upon request.

### 2.4. D. salina UTEX-1644 Strain Transformation

A single colony of *D. salina*, strain UTEX-1644, was inoculated in 150 mL of PKS-modified medium and incubated in a shaker at 150 rpm with a constant light intensity (30 μmol photons m^−2^ s^−1^) provided by fluorescent lamps in a 12/12 light–dark cycle and a temperature of 24 ± 1 °C [30]. Once the culture reached optical density_600nm_ (OD) values of ~0.7 (7 days), 100 mL of the culture was inoculated in 500 mL of PKS-modified medium with magnetic agitation (Corning Pyrex glass Proculture^®^ spinner flasks, Corning, NY, USA). The culture was incubated until it reached OD_600nm_ values of ~0.7 (21 days). Genetic transformation of *D. salina* was performed by co-culture of *D. salina* cells with *A. tumefaciens* as follows: 5 mL of liquid LB culture of *A. tumefaciens* at an OD_600nm_ = 1.0 (20 mg/L rifampicin, 50 mg/L kanamycin) was supplemented with 100 μM acetosyringone and incubated for 4 h; then, the culture was centrifuged at 6000 rpm for 5 min, and the bacterial pellet was resuspended in 5 mL of PKS-modified medium and added to a culture of *D. salina* (500 mL, OD_600nm_ ~0.7). The co-culture was incubated at 25 °C without light for 48 h [34]. The cultures were centrifuged at 1000 rpm for 2 min and washed three times with PKS-modified medium containing 500 mg/L cefotaxime. Subsequently, *D. salina* cells were cultured in PKS-modified medium (50 mg/L hygromycin) and incubated at 25 °C in continuous light for 48 h. To determine the expression of recombinant protein H5rD in *D. salina* cultures, samples were harvested, centrifuged at 13,000 rpm for 15 min, and stored at −80 °C until further analysis.

### 2.5. DNA Extraction and PCR Analysis

To confirm the integration of the H5HA gene named *H5rD*, three samples of the transfected culture of *D. salina* were harvested and analyzed by PCR. Briefly, 10 mL of microalgae cells were centrifuged at 3500 rpm for 5 min, and genomic DNA extraction was performed by the CTAB method [32]. A PCR with specific oligonucleotides targeting the transgene *H5rD* (forward 5′ATGGAAAGAATAGTGATTGCCTTTG3′, reverse 5′TTAGATGCAAATTCTGCACTGC3′) was used to amplify a fragment de 1695 bp. The plasmid pH5HPDS was used as a positive control and DNA of *D. salina* wild type (WT) as a negative control. The cycling conditions were 94 °C for 5 min, 35 cycles at 94 °C for 20 s, 52 °C for 30 s, 72 °C for 180 s, and a final extension at 72 °C for 8 min. PCR products were analyzed by electrophoresis on 1% agarose gel stained with ethidium bromide.

### 2.6. SDS-PAGE of Total Soluble Protein

Samples from total soluble protein (TSP) were obtained from a pellet of 2 g wet weight (WW) of the transfected *D. salina* or from wild-type (WT) *D. salina* cells. To detect H5rD expression, samples were processed as follows: Pellets were resuspended in 10 mL of lysis buffer (1% SDS, 10 mM Tris-MOPS, 2 mM MgCl2, 10 mM KCl pH 7.5, and 2 mM PMSF, added before use). Cell disruption was carried out by sonication at 30% amplitude (10 cycles of 30 s each). Samples were centrifuged at 13,000 rpm for 15 min at 4 °C, and the supernatants were filtered through a polyvinylidene difluoride (PVDF) membrane (pore diameter 0.22 μm, Millex-GV; Millipore, Billerica, MA, USA). Then, samples of each TSP extraction were analyzed with Quick Start Bradford protein assay–Bio-Rad (Bio-Rad Laboratories, Hercules, CA, USA), homogenized TSP concentration 3.5 μg/μL. Finally, samples were analyzed in a 12% SDS-polyacrylamide gel (PAGE): the gel was stained with Coomassie blue, and the level of expression of the H5rD protein was determined using the Quantity One software (Bio-Rad). Each TSP sample was maintained at −80 °C for later use in Western blot (WB) testing and hemagglutinin (HA) assay.

### 2.7. Western Blot (WB) Analysis of H5rD Protein

WB was performed as previously reported [8]. Briefly, 20 μL of each TSP lysates was mixed with protein-loading buffer (10% sodium dodecyl sulfate [SDS], 0.25 M Tris [pH 6.8], 0.1% Bromophenol blue, 7.73% dithiothreitol, and 50% glycerol) in a 1:1 volume and boiled for 5 min, run in a 12% SDS-PAGE gel. Nitrocellulose membranes were incubated with either anti–avian influenza A hemagglutinin antibody (Abcam AB135382, Cambridge, MA, USA, 1:1000 dilution) or anti–H5 AIV antiserum (1:1000 dilution), with a titer using hemagglutination inhibition (HI) test assay of 128 geometric mean titer (GMT) obtained from birds vaccinated with the influenza A-VIREN commercial vaccine (lot 21-078, AIV vaccine; Viren SA de CV, Queretaro, Qro, Mexico). After 12 h of incubation, membranes were washed 2X with TBS buffer, and incubated with either a rabbit anti-mouse IgG H&L (HRP) secondary antibody (Abcam AB97046, Cambridge, MA, USA, 1:2000 dilution) or a rabbit anti-chicken IgY H&L (HRP) antibody (Abcam AB6753, Cambridge, MA, USA, 1:2000 dilution), respectively. Low-pathogenicity AIV strain A/chicken/Hidalgo/28159-232/1994(H5N2) at a dose of 10^3^ 50% chicken embryo infectious dose/0.1 mL was used as a positive control. The virus was treated with NP-40 buffer for protein extraction [35]. A second positive control of the anti–avian influenza A hemagglutinin antibody consisted of a standard protein hemagglutinin HA (Influenza A virus (A/Vietnam/1203/2004(H5N1)) (Abcam AB190125, Cambridge, MA, USA). WB was revealed by incubation for 5 min with a solution of DAB (HRP Color Development Reagent, DAB (3,3′-diaminobenzidine), Bio-Rad Laboratories, Hercules, CA, USA).

### 2.8. Hemagglutination Assay (HA)

According to Killian [36], TSP was evaluated for hemagglutination activity. Briefly, TSP (H5rD protein) in triplicate at an initial concentration of 12.5 µg was two-fold diluted in PBS (pH 7.4) and incubated at 4 °C for two h with 25 μL of 1% chicken erythrocytes in U-bottom 96-well microtiter plates. The highest dilution where complete hemagglutination was observed was considered as one HA unit (HAU). Bovine serum albumin (BSA) with a concentration of 10 µg/mL was used as a negative control. The AIV strain A/chicken/Hidalgo/28159-232/1994(H5N2) was used as a positive control [37].

## 3. Results

### 3.1. Cloning of H5HA and Expression of Recombinant Protein H5rD in D. salina

The correct integration of the H5HA gene was confirmed by three randomly selected samples of the purified genomic DNA from transfected cultures of *D. salina*. A 1695 bp amplicon was observed in all three samples corresponding to the transgene *H5rD*. No amplification was observed in the genomic DNA from *D. salina* WT (Figure 1a).

The expression of a protein from TSP was confirmed in *D. salina* pellets from the transfected cells (Figure 1b). Results from a single sample showed the presence of a protein of approximately molecular weight (MW) 69 kDa, the putative recombinant protein H5rD. According to the quantitative densitometry of proteins stained with Coomassie blue, approximately 255.5 μg of recombinant protein was recovered from 2 g WW of *D. salina*. The expression of H5rD was confirmed by the TSP from *D. salina* either with monoclonal antibodies (Figure 1c) or with the anti-H5 AIV serum (Figure 1d). Samples from *D. salina* WT showed no antibody reactivity. The recombinant protein H5rD has a length of 564 corresponding to the amino acid sequence reported in GenBank # ABB88379.1 for the strain A/chicken/Hidalgo/28159-232/1994(H5N2).

### 3.2. Bioactivity Analysis

The activity of the protein H5rD as a native H5HA was determined by HA assay. To increase the concentration of recombinant protein H5rD, a sample of 1 mL of TSP from transfected *D. salina* was concentrated by ultra-filtering, and upon 10-fold increase in the sample, a sufficient concentration was achieved. The hemagglutination results show that for all three samples of protein H5rD, complete hemagglutination was observed at a 1:128 dilution, corresponding to 1 HAU (Figure 1e, rows 1–3). H5rD protein concentrations of ≥0.09 µg were unable to induce hemagglutination. No hemagglutination activity was observed in wells with a negative control (Figure 1e, row 4); the positive control was a HA titer of 1:1024.

## 4. Discussion

Nowadays, recombinant antigen production systems are more economically viable [38]. Innovative alternative platforms for the generation of antigens for disease research and control are required around the world [39]. In the case of AI, producing viruses in biological models such as embryos is an essential part of the vaccine development process [40,41]. However, embryos represent a high cost, and they must be used on animals in the manufacturing stage [42]. As a result, there is growing interest at the laboratory and pharmaceutical levels in developing alternative antigen manufacturing methods.

In this study, we have shown the successful expression of soluble H5HA protein from avian influenza subtype H5N2 in a microalgae system using *D. salina* as a host species. HA protein expression has previously been reported in bacteria, plants, insect cells, mammalian cell cultures, and microalgae, referring to different levels of expression and glycosylation efficiency [8,12,43], but no advantages of using *D. salina* as a protein expression system for IAV have been identified.

Employing biological systems to express recombinant proteins is required to fulfil several properties, including levels of expression, protein complexity, and glycosylation processes [13]. Proper protein folding and functionality, along with glycosylation, are the most critical characteristics to achieve [44]. Therefore, based on the results of WB and reactivity to monoclonal and polyclonal antibodies directed against the H5HA, the findings suggest the proper expression of the H5rD protein.

Recent research on microalgae has shown that these unicellular organisms have benefits over other systems [45]. There are several distinct advantages compared to other bioreactor systems, including: biomass doubling in 24 h; relatively short growth times; the expression of proteins in the nucleus, chloroplast, and mitochondria; post-translational modifications; phototrophic or heterotrophic growth; temperature-controlled conditions, light, and nutrients; reduced risk of escape of transgenes into the environment; some species being recognized as GRAS (Generally Recognized as Safe for humans); and, finally, the possibility of their lyophilization being used for storage or for oral administration [13,46,47]. Using *D. salina*, we report that in a timeframe of 30 days, it was possible to recover up to 225 μg of soluble protein from 2 g of WW of a culture of *D. salina*. These results are encouraging considering that the actual process followed to obtain a vaccine strain takes no less than six months and requires an average of 100 embryonated eggs for 10 mg of total viral protein [40], as well as causes possible contamination with AIV in the embryo-producing flocks, putting at risk the availability of these flocks for the production of vaccines [48].

In prior research using *D. salina*, the expression levels of recombinant proteins were observed at similar levels [49]. In models of transitory expression of HA protein in plant tissue, they can reach up to 9.7% of the TPS [50], and when excreted into the medium they can reach up to 1 mg/50 mL of HAr in the case of *Schizochytrium sp.* [51]. However, because a considerable proportion of recombinant HA protein in plant tissue is found in the insoluble protein, the amount of H5rD in *D. saline* is likely to be greater than that which is observed in TPS [51]. The cultivation time, medium cost, and contamination management we established in our study underline *D. salina* advantages.

In the case of post-translational modifications such as glycosylation in proteins, the microalgae system showed the capacity to carry out these modifications, allowing the functionality of complex proteins with biological activity, as is the case for the HA protein [51]; models such as *Chlamydomona reinhardtii* and *Schizochytrium sp.*, among others [51,52], have been shown to fulfil the characteristics necessary to achieve the expression of recombinant proteins with very specific glycosylation requirements [53]. These characteristics, together with studies of the metabolic pathways and associated genes in processes such as glycosylation, demonstrate that these organisms can generate post-translational modifications to achieve the folding, functionality, and antigenicity required in each case [54]. However, more studies are required in models such as *D. salina* to determine these modifications, their possible effects on the characteristics of the proteins expressed in this system, and their interaction in an animal model.

Among the main characteristics of the HA protein needed for its correct folding and the presence of hemagglutinating activity (viral receptor) is post-translational modifications glycosylation [53]. In the case of HA, glycosylation patterns vary between the different strains and types of HA [53]. Even the presence of N-glycosylation sequences is not sufficient to be glycosylated; in addition to this, the type of cell where the virion replicates affects these post-translational modifications (chicken embryo, human cells, etc.) [55]. Glycosylation is constantly present where different types of HA (glycosides 27, 40, 176, 303, and 497) are recognized. However, 27 is necessary to acquire biological function [53]. Since the receptor-ligand activity of the HA protein requires post-translational modification such as 27 N-glycosylation, the determination of biological activity of the H5rD protein using hemagglutination assay allows the demonstration of a functional protein that requires post-translational modification. This work shows accurate production of the H5rD protein and biological activity, where HA titers were comparable to the wild type of virus used as a control. The receptor binding capacity of the H5rD protein is similar to that which was previously reported in microalgae models [50]; hence, its expression in this biological model can be regarded as viable.

Our study has some limitations; further studies are needed to evaluate if the H5rD protein is expressed as native HA protein, affecting the antigenicity [56] and the possibility of induction of an immune response in an animal model. Moreover, further studies are needed to confirm the presence of post-translation modification as a factor that could also affect antigenicity and protein folding [57,58].

## 5. Conclusions

According to the current findings, the *D. salina* system can carry out the expression and accurate folding of HA protein, allowing the biochemical features and biological activity of binding to chicken erythrocytes to be preserved. As a result, the *D. salina* expression system is a viable option for the creation of novel recombinant protein production methodologies.

## Figures and Tables

**Figure 1 microorganisms-10-00361-f001:**
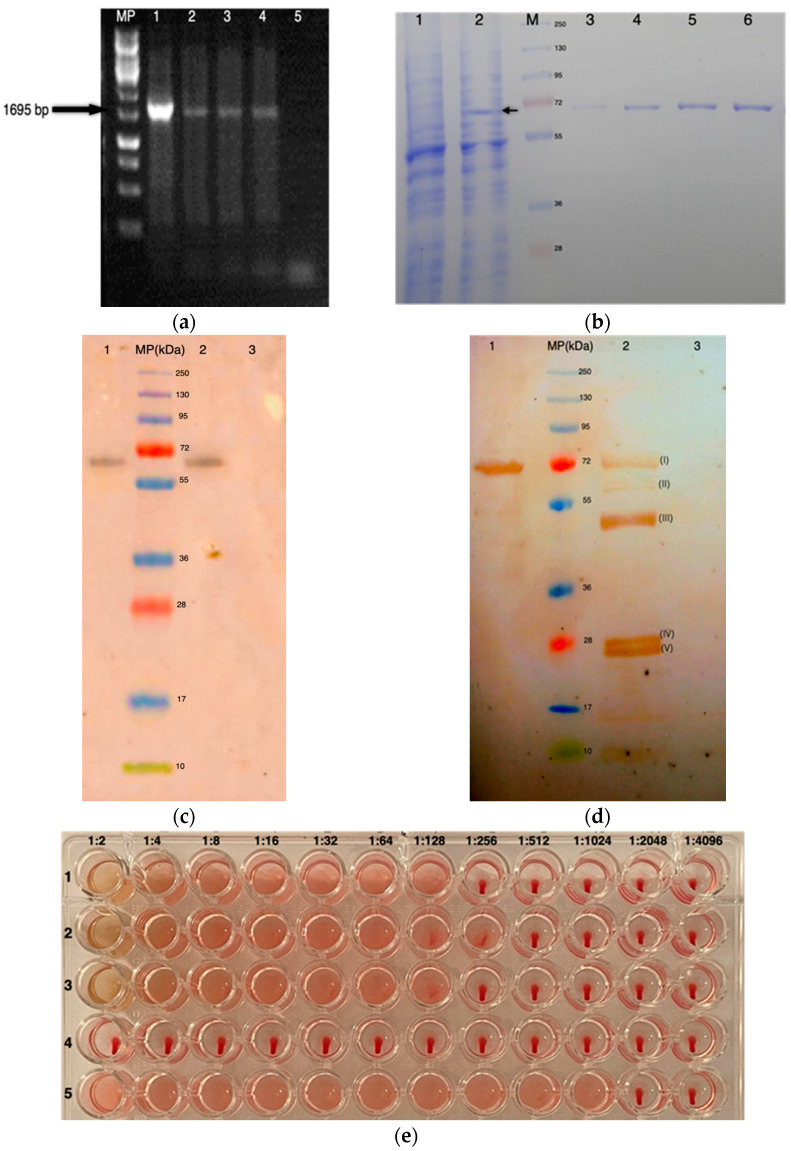
(**a**) One percent agarose gel to show PCR products corresponding to the *H5HA* gene of 1695 bp: lane 1, positive control (plasmid DNA with the gene *H5rD*); lanes 2–4, PCR product from 3 samples from transformed *D. salina* culture; lane 5, negative control (DNA from 1 culture sample of untransformed *D. salina*); MP, molecular weight marker (GeneRuler 1 kb DNA Ladder SM0311, Thermo Fisher, Waltham, MA, USA). (**b**) Coomassie blue-stained 12% SDS-PAGE gel showing the expression of a putative protein in transformed *D. salina* cultures: lane 1, TSP of WT *D. salina* cultures; lane 2, TSP of transformed *D. salina* cultures (adjusted to 70 μg); lane MP, molecular weight marker (PageRuler™ Plus Prestained Protein Ladder, 10 to 250 kDa 26619, Thermo Fisher, Waltham, MA, USA); lanes 3–6, BSA at total concentration of 0.1, 0.5, 1, and 1.5 μg, respectively. (**c**) Western blot of the H5rD protein with monoclonal antibodies: lane 1, H5rD protein detected with anti–avian influenza A hemagglutinin antibody (Abcam AB135382, Cambridge, MA, USA); lane 2, H5 standard protein as positive control (Abcam AB190125, Cambridge, MA, USA) molecular weight ~64 kDa without post-translational modification; lane 3, TSP sample from *D. salina* WT; MP, molecular weight marker (PageRuler™ Plus Prestained Protein Ladder, 10 to 250 kDa 26619, Thermo Fisher, Waltham, MA, USA). (**d**) Western blot detection of H5rD protein with polyclonal antibodies: lane 1, H5rD protein from TSP detected with a chicken serum (IgY anti-H5); lane 2, viral proteins from low pathogenic virus A/Chicken/México/232/94/CPA; lane 3, TSP sample from *D. salina* WT; lane MP, molecular weight marker (PageRuler™ Plus Prestained Protein Ladder, 10 to 250 kDa 26619, Thermo Fisher, Waltham, MA, USA); protein bands identified in lane 2 correspond to (I) HA, (II) NA / NP, (III) HA1, (IV) M1 and (V) HA2. (**e**) HA assay of ultrafiltered H5rD protein incubated with 1% suspension of chicken erythrocytes: row 1–3, Microalgae-produced hemagglutinin; row 4, BSA negative control serially diluted, initial concentration of 10 µg/mL; row 5, allantoic fluid from chicken embryos infected with low pathogenic virus A/Chicken/México/232/94/CPA as a positive control.

## Data Availability

Not applicable.

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
