# Peer review of "Transformation of Dunaliella salina by Agrobacterium tumefaciens for the Expression of the Hemagglutinin of Avian Influenza Virus H5"

_microorganisms, 2022, doi:10.3390/microorganisms10020361_

Round 1

Reviewer 1 Report

The introduction needs to have some explanation as to what the investigators have used the algal approach? So how does the algal system offer advantages over a yeast expression system for example? The yeast system has been used for a long time and validated. This is covered in part in the first paragraph in the discussion- but the authors do need to have some explanation for the necessity of the approach even if it is purely from a vaccine production viewpoint.

Line 86 PKS should be written in full

Line 295-300  So the immunoreactivity of the recombinant H5d suggests that the protein is folded correctly to preserve the epitopes but what about the glycosylation  of the recombinant protein? Is there not a simple test that could be used to confirm rather than just speculate that pot translation glycosylation is correct with the recombinant protein. This result would provide a level of validation for the choice of using the algal system.

Author Response

Dear Reviewer, #1, thank you very much for the time you have spent reviewing our manuscript. Your comments are very valuable and helpful for revising our paper and guiding our research. We have studied those comments carefully and have made corrections, which we hope to meet with the approval. English language, style and spell check have been revised. The reviewed portion in the new version was included and are highlighted in yellow in the reviewed manuscript. The following is our point-by-point response to reviewers’ comments:

“The introduction needs to have some explanation as to what the investigators have used the algal approach? So how does the algal system offer advantages over a yeast expression system for example? The yeast system has been used for a long time and validated. This is covered in part in the first paragraph of the discussion- but the authors do need to have some explanation for the necessity of the approach even if it is purely from a vaccine production viewpoint. (Describe the principal difference between yeast and Dunaliella)"

Suggestion accepted. A new paragraph has been included in the introduction.  Thank you

"Line 86 PKS phosphate potassium sodium should be written in full"

Suggestion accepted. Phosphate potassium sodium has been written in full.  Thank you

"Line 295-300  So the immunoreactivity of the recombinant H5d suggests that the protein is folded correctly to preserve the epitopes but what about the glycosylation of the recombinant protein? Is there not a simple test that could be used to confirm rather than just speculate that pot translation glycosylation is correct with the recombinant protein. This result would provide a level of validation for the choice of using the algal system." 

We appreciate your comments.  In this regard, previous transformation assays with other microalgae models have demonstrated both the folding capacity and the presence of biological activity of complex proteins such as HA of the influenza virus; Despite not performing a test to determine the presence of post-translational modifications, by demonstrating the ligand-receptor activity in the HA assay, the formation of a functional structure is demonstrated, which requires such post-translational modifications. In addition to this data, it is important to mention that this test forms the first part of a project that includes the subsequent analysis of these proteins in an animal model, for the determination of antigenicity, as well as more biochemical tests required to corroborate their characteristics and feasibility of use as an antigen for viral challenge tests in commercial birds. Thank you.

Reviewer 2 Report

The manuscript ‘Transformation of Dunaliella salina by Agrobacterium tumefaciens for the expression of the hemagglutinin of avian influenza virus H5’ by Castellanos-Huerta et al. describes new host system for expression of influenza hemagglutinins. The work is concise and well described. However, I have three major comments that are to be addressed before the publication.

  • There is no native PAGE that allows estimation of trimeric HA. This simple experiment is necessary for the characterization of the protein
  • If the experimental Mw is the same as Mw predicted for the unglycosylated protein, the glycosylation did not occur. Glycosylation is to be proved in some way. For example, the protein can be deglycosylated by some glycosidase that can be seen in SDS-PAGE due to the decrease in Mw. Human cells provides nearly 10 kDa of oligosaccharide per HA moiety (for example, see https://www.abcam.com/recombinant-influenza-a-virus-hemagglutinin-h3-protein-his-tag-ab217661.html)
  • Algae can provide N-glycosylation but the oligosaccharides differ from human ones. I refer to two reviews: doi 3389/fpls.2020.609993 and doi 10.3389/fpls.2014.00359. The latest review showed that D. salina does not provide N-glycosylation at all. The discussion of these data is necessary both for the reader and for careful interpretation of the results

Author Response

Dear Reviewer, #2, thank you very much for the time you have spent reviewing our manuscript. Your comments are very valuable and helpful for revising our paper and guiding our research. We have studied those comments carefully and have made corrections, which we hope to meet with the approval. English language, style and spell check have been revised. The reviewed portion in the new version was included and are highlighted in yellow in the reviewed manuscript. The following is our point-by-point response to reviewers’ comments:

“The manuscript ‘Transformation of Dunaliella salina by Agrobacterium tumefaciens for the expression of the hemagglutinin of avian influenza virus H5’ by Castellanos-Huerta et al. describes a new host system for the expression of influenza hemagglutinins. The work is concise and well described. However, I have three major comments that are to be addressed before the publication.”

“There is no native PAGE that allows the estimation of trimeric HA. This simple experiment is necessary for the characterization of the protein”

We appreciate your comment.  In this regard, previous transformation assays with other microalgae models have demonstrated both the folding capacity and the presence of biological activity of complex proteins such as HA of the influenza virus; Despite not performing a test to determine the presence of trimeric HA structure, by demonstrating the ligand-receptor activity in the HA assay, the formation of a functional structure is demonstrated, which requires such complex structure; since the subunits are not functional for HA test. In addition to this data, it is important to mention that this test forms the first part of a project that includes the subsequent analysis of these proteins in an animal model, for the determination of antigenicity, as well as more biochemical tests required to corroborate their characteristics and feasibility of use as an antigen for viral challenge tests in commercial birds. the results in the animal model test will provide more detailed characteristics of H5rD protein, because the immune response is dependent on the structure and complexity of the HA protein, as previously demonstrated in subunit antigens (bacteria origin)

“If the experimental Mw is the same as Mw predicted for the unglycosylated protein, the glycosylation did not occur. Glycosylation is to be proved in some way. For example, the protein can be deglycosylated by some glycosidase that can be seen in SDS-PAGE due to the decrease in Mw. Human cells provide nearly 10 kDa of oligosaccharide per HA moiety (for example, see https://www.abcam.com/recombinant-influenza-a-virus-hemagglutinin-h3-protein-his-tag-ab217661.html)”

We appreciate your comments. According to what was observed by the WB test, it is possible to show that the H5rD protein has an approximate weight of 69 kDa, the weight described in the results will be corrected (64 kDa will be substituted), this is due to the determination by means of the distance between the molecular weight marker and the H5rD protein. It is important to mention that both the native protein in the polyclonal WB and the recombinant protein of Baculovirus infected Sf9 cells origin in the monoclonal WB assay, are located at a similar height as the H5rD protein, so it can be deduced that their weight is what would be expected when carrying out post-translational modifications for these proteins, which is higher than 64 kDa in both cases (Fig 1b, 1c and 1d).

“Algae can provide N-glycosylation but the oligosaccharides differ from human ones. I refer to two reviews: doi 3389/fpls.2020.609993 and doi 10.3389/fpls.2014.00359. The latest review showed that D. salina does not provide N-glycosylation at all. The discussion of these data is necessary both for the reader and for careful interpretation of the results”

Suggestion accepted. A new paragraph has been included in the discussion section for better clarification.  Thank you

Round 2

Reviewer 2 Report

The manuscript was not supplied with additional experimental data on glycosylation and oligomeric state of the protein. I suggest rejection of the manuscript as the experimental part is not complete.